



# Elemental stoichiometry and Rock-Eval® thermal stability of organic matter in French topsoils

Amicie A. Delahaie[1], Pierre Barré[1], François Baudin[2], Dominique Arrouays[3], Antonio Bispo[3], Line Boulonne[3], Claire Chenu[4], Claudy Jolivet[3], Manuel P. Martin[3], Céline Ratié[3], Nicolas P. A. Saby[3],
Florence Savignac[2], Lauric Cécillon[1]

[1] Laboratoire de Géologie, École normale supérieure, CNRS, PSL Univ., IPSL, Paris, France
[2] UMR 7193 ISTeP, Sorbonne Université, Paris, France
[3] INRAE, US1106, InfoSol, F-45075 Orléans, France
[4] UMR ECOSYS, Université Paris Saclay, INRAE, AgroParisTech, 91120 Palaiseau, France

*Correspondence to*: Amicie A. Delahaie (amicie.delahaie@ens.fr) and Pierre Barré (barre@geologie.ens.fr). Laboratoire de Géologie de l'École Normale Supérieure (UMR 8538), 24 Rue Lhomond, 75005 Paris, France.

**Abstract.** The quality and quantity of soil organic matter (SOM) are key elements of soil health and climate regulation by soils. The Rock-Eval® thermal analysis technique is increasingly used as it represents a powerful method for SOM
characterization by providing insights on bulk SOM chemistry and thermal stability. In this study, we applied this technique on a large soil sample set from the first campaign (2000–2009) of the French monitoring network of soil quality: RMQS. Based on our analyses on ca. 2000 composite surface (0–30 cm) samples taken all over mainland France, we observed a significant impact of land cover on both SOM thermal stability and elemental stoichiometry. Cropland soils had a lower mean value of hydrogen index (a proxy for SOM H/C ratio) and a higher thermal stability than grasslands and forests.
Regarding the oxygen index (a proxy for SOM O/C ratio), we observed significant differences in values for croplands, grasslands and forests. Positive correlations between the temperature parameters on the one hand and the clay content and pH on the other hand highlight the protective effect of clay on organic matter and the impact of pH on microorganisms mineralization activity. Surprisingly, we found weak effects of climatic parameters on the thermal stability and stoichiometry of SOM. Our data suggest that topsoil SOM is on average more oxidized and biogeochemically stable in croplands. More
generally, the high number and even repartition of data on the whole French territory allow to build a national interpretative referential for these indicators in surface soils.

## 1 Introduction

The fate of soil organic carbon (SOC) is crucial from both soil health and climatic perspectives. In terms of soil health, SOC plays an important functional role. Its decomposition by microorganisms provides energy to the whole soil food web and key
nutrients to plants and soil fauna. SOC also regulates the water cycle through controlling soil structure (Rawls et al., 2003). From a climatic perspective, soils can act both as a source or a sink of carbon (Amundson, 2001; Eglin et al., 2010).



Maintaining or increasing SOC stocks has become a key policy issue for the coming decades (Rumpel et al., 2018) that raises a number of important scientific challenges regarding our knowledge of SOC dynamics (Dignac et al., 2017).

The evolution of SOC stocks depends on the balance between soil carbon inputs (mostly by plants) and outputs (mostly by
microbial decomposition). The persistence of SOC determines soil carbon outputs so that estimating the biogeochemical stability of SOC to microbial decomposition (i.e. the difficulty for microorganisms to mineralize SOC) is of paramount importance to infer SOC dynamics (Schmidt et al., 2011; Lehmann and Kleber, 2015). Indeed, a better knowledge of SOC persistence would allow refining our estimates of the soil carbon inputs needed to maintain or enhance SOC stocks. However, estimating the biogeochemical stability of SOC is a challenging task because its turnover time encompasses a
broad spectrum (ranging from some days or weeks to centuries; Balesdent and Guillet, 1982) resulting from a series of interacting SOC stabilization mechanisms. Indeed, SOC can be protected from microbial decomposition due to its chemical nature (e.g., pyrogenic SOC), its interactions with soil mineral surfaces or its spatial inaccessibility for microbes (Baldock and Skjemstad, 2000; Von Lützow et al., 2006).

Several routine techniques have been proposed to separate fractions that are labile, intermediate or stable at various
timescales (von Lützow et al., 2007, Bispo et al., 2017; Chenu et al., 2015). However, none of these techniques manages to isolate precisely homogeneous fractions with the same biogeochemical stability (von Lützow et al., 2007; Poeplau et al., 2018; Cécillon et al., 2021). Common methods include biological respiration measurements performed during laboratory incubations of soils (e.g. Collins et al., 2000) and various physical (particle size or density) and/or chemical (aqueous or organic extraction) SOC fractionation methods (von Lützow et al., 2007).

Thermal analysis methods have been used for several decades to study the characteristics of soil organic matter (SOM). Many different methods exist, which measure different variables: e.g. thermogravimetry, differential scanning calorimetry, evolved gas analysis, etc. (Plante et al., 2009). A multitude of variations in temperature ramps, compositions of reaction atmosphere and measured parameters are encountered within each class of methods. Some thermal analysis methods provide indicators that are related to SOM biogeochemical stability: the more biogeochemically stable SOM is, the more thermally
stable and energy- and hydrogen-depleted it is (Barré et al., 2016; Sanderman and Grandy, 2020). Among thermal analysis methods, Rock-Eval® thermal analysis is increasingly used to derive thermal indicators related to SOC biogeochemical stability (Gregorich et al., 2015; Saenger et al., 2015; Cécillon et al., 2018; 2021; Poeplau et al., 2019; Chassé et al., 2021).

The Rock-Eval® method was developed in the 1970s. Initially intended for the characterization of petroleum source rocks and sediments in order to estimate their potential for hydrocarbon extraction (Espitalié et al., 1977), this method was then
adapted to the study of SOM (Disnar et al., 2003). This technique allows the measurement of the organic and inorganic carbon content of a soil sample, as well as numerous indicators of the thermal stability and elemental stoichiometry of SOM. Espitalié et al. (1977) has shown that the Rock-Eval® hydrogen index HI (respectively the oxygen index OIre6) is a good proxy for the H/C ratio (respectively O/C ratio) of organic matter in kerogens and, later, soils. Many temperature parameters can also be calculated to give insights on the thermal stability of SOM (Gregorich et al., 2015; Sebag et al., 2016; Cécillon et
al., 2018). With a rate of one sample per hour at a reasonable price (below 50 USD per sample), Rock-Eval® thermal





analysis is a particularly fast and simple tool to use, and is therefore well suited to the study of large soil sample sets, such as the ones collected in the context of national or continental soil monitoring networks.

However, so far, the different existing soil monitoring networks worldwide have not used thermal analysis methods to infer SOC biogeochemical stability. Some of them have focused on SOC physical fractionation schemes, in combination with

infrared spectroscopy or environmental variables (e.g. Vos et al., 2018; Viscarra-Rossel et al., 2019; Lugato et al., 2021; Sanderman et al., 2021). Here, we used Rock-Eval® thermal analysis to investigate the thermal stability and elemental stoichiometry of topsoil samples of the first campaign of the French national soil monitoring network (RMQS, Réseau de Mesures de la Qualité des Sols; GIS Sol; https://www.gissol.fr/le-gis/programmes/rmqs-34). The RMQS network has been designed for the long-term monitoring of the soil quality of the whole French territory by collecting information and

sampling soils on, every 15 years in average, a set of 2170 sites at the locations of a regular, square grid thus forming a systematic sample (Jolivet et al., 2006; English version to be available online). The first campaign took place between 2000 and 2009 in mainland France, covering 7 major land cover types (croplands, grasslands, forests, vineyards & orchards, wastelands, poorly human-disturbed environments, and gardens).

In this study, we aimed in the first place to verify that the Rock-Eval® method was suited to characterize SOM on archived

soil samples at the scale of a monitoring network. For this purpose, we checked if the organic and inorganic carbon yields of the Rock-Eval® thermal analysis for soil samples, calculated by comparing Rock-Eval® estimates to reference methods, were acceptable. Second, we computed several common Rock-Eval®-based indicators in order to perform an unprecedented country-wide evaluation of the thermal stability and elemental composition of the SOM. Third, thanks to the numerous environmental data available at each RMQS site, we aimed at studying the relationships between land-cover, climate and soil

properties and the SOC-related indicators derived from Rock-Eval® thermal analysis.

## 2 Material and methods

### 2.1 Topsoil sampling and processing

A full description of the RMQS and of the soil sampling process of its first sampling campaign is available in Jolivet et al., 2006. Briefly, the soil is monitored at the locations of a regular, square grid with a resolution of 16 km. A sampling site was

settled when possible at the center of the cell; otherwise, an alternative site was taken within a 1 km radius from the center of the cell. This resulted in a total of 2170 RMQS sites in mainland France. At each selected site, 25 topsoil samples (0–30 cm or tilled layer depths) were taken with a spiral soil auger from a 20 m $\times$ 20 m sampling area then mixed to provide a composite sample. Subsoil samples were also taken, but were not considered in the present study.

The composite samples (5 to 10 kg of bulk soil) were air-dried at 30°C in trays for 8 to 10 days on average. The samples

were then quartered according to NF ISO 11464 to obtain a sub-sample of ca. 650 g. They were then crushed by hand to break aggregates while preserving calcareous and/or ferro-manganic nodules and sieved at 2 mm. The remains of the



composite samples were stored in water-tight plastic buckets. An aliquot of each air-dried and sieved composite sample was then finely ground using a Cyclotec 1093 (Foss).

Of the 2170 archived aliquots of finely ground topsoil samples from the first RMQS sampling campaign in mainland France,
2037 were recovered and used for this study. When necessary, the samples were manually ground again using an agate mortar to reach the particle size requirements of Rock-Eval® thermal analysis of soils (below ca. 250 µm).

## 2.2 Physical and chemical soil analyses

The physical and chemical soil analyses were carried out on the 2 mm sieved composite samples at the Laboratoire d'Analyse des Sols (INRAE, Arras, France). Among the large set of soil properties measured, we selected in this study the
following ones (Jolivet et al., 2006): particle-size measurements without decarbonation in g.kg⁻¹ of sample (Robinson pipette and underwater sieving, method validated in relation to standard NF X31-107) leading to five fractions (clay: ≤ 2 µm; fine silt: 2–20 µm; coarse silt: 20–50 µm; fine sand: 50–200 µm; coarse sand: 200–2000 µm); pH in water (NF ISO 10390, dilution with ⅕); total carbonate content in g.kg⁻¹ of sample (volumetric method NF EN ISO 10693) to estimate the total inorganic carbon $Cinorg$ = Total carbonate $\times$ 0.12 in g.kg⁻¹ of sample; total carbon content (g.kg⁻¹ of sample) determined by
elemental analysis using dry combustion on non-decarbonated soil; organic carbon content derived from the elemental analysis (TOCea; g.kg⁻¹ of sample) calculated as Total carbon - Cinorg (NF ISO 10694); total nitrogen in g.kg⁻¹ of sample (dry combustion NF ISO 13878); CEC in cmol+.kg⁻¹ of sample (cobaltihexammine chloride extraction NF X31-130); free iron oxides in g/100 g measured with the Tamm method in the dark and Mehra-Jackson method (INRA standard/NF ISO 22036).

## 2.3 Rock-Eval® thermal analysis

### 2.3.1 Thermal analysis process

Rock-Eval® thermal analyses were carried out at the UMR 7193 ISTeP (Sorbonne Université, Paris, France) on the 2037 recovered samples according to the routine classically used for soil samples (Disnar et al., 2003; Baudin et al., 2015). Approximately 60 mg of each finely ground topsoil sample was used for the Rock-Eval® thermal analysis on a RE6 turbo
device (Vinci Technologies, France). For each analysis, the sample was placed in a special high-temperature resistant stainless steel pod allowing the transport gas to pass through. It first underwent a pyrolysis step under an inert $N_2$ atmosphere. After a three-minute isotherm at 200 °C, the sample was heated to 650 °C following a temperature ramp of 30 °C/min. The flame ionization detector (FID) monitored the gaseous emissions of carbon from hydrocarbon compounds (HC_PYR Rock-Eval® thermogram), while CO (CO_PYR Rock-Eval® thermogram) and $CO_2$ (CO₂_PYR Rock-Eval®
thermogram) were detected by an infrared detector. The second step is an oxidation (laboratory air atmosphere with $CO_2$ and $H_2O$ previously removed, i.e., in presence of oxygen): the sample experienced a one-minute isotherm at 300 °C, then was raised to 850 °C following a 20 °C/min ramp, and finally remained on a five-minutes isotherm at 850 °C. The evolution of





CO and $CO_2$ were again monitored using the infrared detector during the oxidation phase (CO_OX and $CO_2$_OX Rock-Eval® thermograms). The resulting five thermograms were processed using the Geoworks software (Vinci Technologies, Geoworks V1.6R2), except for the parameters R-index, I-index which were computed using homemade Python scripts according to the formula proposed by Sebag et al. (2016).

Our Rock-Eval® thermal analyses campaign included duplicated soil analyses (one every eight samples), performed in order to check the reproducibility of the analyses, along with standard analyses (one every nine samples) to check the calibration of the device and identify a possible drift in the analysis. The Rock-Eval® thermal analysis of a soil sample measures its total organic carbon (TOCre6) and total inorganic C (MinC) contents that sum to total carbon content (see Behar et al., 2001 for a detailed description). The organic carbon yield of Rock-Eval® thermal analysis was defined as TOCre6/TOCea, while its inorganic carbon yield was defined as MinC/Cinorg, and its total carbon yield was defined as (TOCre6+MinC)/(TOCea+Cinorg). We used the organic carbon yield of Rock-Eval® thermal analysis to select soil samples among duplicates: only the one with the best yield was conserved. When assessing SOM thermal stability and elemental stoichiometry, it is essential to ensure that SOM analyzed by the thermal analysis method corresponds to SOM measured using the reference elemental analysis method. We therefore proposed a quality criterion for Rock-Eval® thermal analysis based on its organic carbon yield, with an arbitrary acceptable range of yields from 0.7 to 1.3.

### 2.3.2 Rock-Eval® parameters

Many usual Rock-Eval® parameters were calculated from the thermograms (Table A). First there are parameters related to carbon quantities : the total organic carbon (TOCre6; $g.kg^{-1}$ sample); the total inorganic carbon (MinC; $g.kg^{-1}$ sample); the amount of pyrolyzable organic carbon (PC; $g.kg^{-1}$ sample); the ratio of pyrolyzable organic carbon over total organic carbon (PC/TOCre6; no unit); the carbon released during the first pyrolysis isotherm (PseudoS1; $g.kg^{-1}$ sample); the carbon released as hydrocarbons during the pyrolysis except during the first isotherm (S2; $g.kg^{-1}$ sample); the ratio of carbon released as hydrocarbons during the pyrolysis except during the first isotherm over the pyrolyzable organic carbon (S2/PC; no unit). Second, there are temperature parameters related to the SOC thermal stability. Their calculation was performed over different intervals of integration depending on the thermogram. The upper limits of the integration ranges were selected to exclude CO and CO2 signals derived from carbonates. The temperature parameter T50_HC_PYR (respectively T70_HC_PYR and T90_HC_PYR; °C) is defined as the temperature at which 50 % (respectively 70 % and 90 %) of the hydrocarbon effluents have been emitted during the pyrolysis ramp (the initial isotherm is excluded; the integration ends at 650 °C). Similarly, T30_CO2_PYR (respectively T50_CO2_PYR, T70_CO2_PYR and T90_CO2_PYR; °C) is the temperature at which 30 % (respectively 50 %, 70 % and 90 %) of the $CO_2$ have been emitted during the pyrolysis ramp (the beginning isotherm is excluded; the integration ends at 560 ° C); T50_CO_PYR (°C) is the temperature at which 50 % of the CO have been emitted during the pyrolysis ramp (the beginning isotherm is excluded; the integration ends at 560 °C). T50_CO2_OX (respectively T70_CO2_OX and T90_CO2_OX; °C) is the temperature at which 50 % (respectively 70 % and 90 %) of the $CO_2$ have been emitted during the oxidation phase (the integration ends at 611 °C); T50_CO_OX



(respectively T70_CO_OX; °C) is the temperature at which 50 % (respectively 70 %) of the CO have been emitted during the oxidation phase (the integration ends at 850 °C). We also calculated two other parameters previously used in assessing the thermal stability of SOC: the I-index (related to the thermolabile organic carbon released as hydrocarbon effluents, Sebag et al., 2016; no unit) and the R-index (the proportion of thermostable organic carbon released as hydrocarbon effluents after

400 °C, Sebag et al., 2016; no unit). Finally, we calculated the three following Rock-Eval® parameters, related to the SOM stoichiometry. The Hydrogen Index, HI, is the ratio of emitted hydrocarbons to TOCre6 (unit g HC.kg$^{-1}$ TOCre6); it is calculated following Eq. (1):

$$HI = \frac{S2 \times 100}{TOCre6} \tag{1}$$

where S2 is the hydrocarbons signal during pyrolysis (Behar et al., 2001). The Oxygen Index, OIre6, is the ratio of organic

oxygen to TOCre6 (unit g $O_2$.kg$^{-1}$ TOCre6); it is calculated following Eq. (2):

$$OIre6 = \frac{16}{28} \times \frac{S3CO \times 100}{TOCre6} + \frac{32}{44} \times \frac{S3 \times 100}{TOCre6} \tag{2}$$

where S3 and S3CO are respectively the organic $CO_2$ and organic CO signals during pyrolysis (Behar et al., 2001; Cécillon et al., 2018). The ratio of hydrogen amount to oxygen amount is HI/OIre6 (no unit).

As presented above, the treatment of the five thermograms can result in the production of a multitude of Rock-Eval®

parameters. We have decided to present the results on the following parameters in more detail: T50_HC_PYR; T90_HC_PYR; T50_CO2_PYR; T50_CO2_OX; I-index; R-index; HI and OIre6. The results obtained for some other Rock-Eval® parameters are presented as Supplementary Information. The temperature parameters T90_HC_PYR, T50_CO2_PYR and T50_CO2_OX were selected as they are derived from the 3 different thermograms contributing the most to the Rock-Eval® signals, and as they are well correlated to the proportion of centennially stable SOC in temperate soils (Cécillon et al.,

2021) and were used in some previous studies (e.g. Barré et al., 2016; Poeplau et al., 2019). The parameters T50_HC_PYR, I-index and R-index were selected as they were used in several previous studies (e.g. Gregorich et al., 2015; Sebag et al., 2016; Matteodo et al., 2018; Soucémarianadin et al., 2018). HI and OIre6 were selected as they are usual Rock-Eval® parameters and they give insights on the elemental stoichiometry of SOM.

## 2.4 Climate data

Climate data were extracted from the French SAFRAN database (https://publitheque.meteo.fr/okapi/accueil/okapiWebPubli/index.jsp). The daily data were averaged over the 1969-1999 period in order to compute for each site the mean annual temperature (MAT) and mean annual precipitation (MAP).

## 2.5 Statistical analysis

We calculated linear regressions without intercept using the measurements of the organic, inorganic and total carbon yield of

Rock-Eval® thermal analysis to verify the ability of the Rock-Eval® thermal analysis to accurately measure the carbon amount of the samples.





To assess the effect of land cover on the Rock-Eval® parameters, we performed pairwise comparisons of medians by non-parametric Kruskal–Wallis tests ($P < 0.05$) followed by Wilcoxon tests, with $P < 0.05$ for each pair. The correction of p-values in the framework of the multiple comparisons was done with the Holm-Bonferroni method. Correlations between parameters were calculated using the Spearman method. We conducted a principal component analysis (using the R library FactoMineR) using all the observations and 11 pedoclimatic parameters: clay, total silt and total sand contents, pH in water, residual water content, carbonate content, mean annual temperature, mean annual precipitation, Tamm and Mehra-Jackson iron oxyhydroxides contents and C/N ratio (Fig. D). The data processing and statistical analysis were carried out with the R software (V4.1.2; R Core Team 2021. Packages integrated to R: base, datasets, graphics, grDevices, methods, stats, utils. Packages added: corrplot, car, ggplot2, ggpubr, factoextra, plot3D, rstatix, sf, tmap). The point maps of the Rock-Eval® Hydrogen Index and T50_CO2_PYR values were obtained using the tmap and sf R packages.

## 3 Results

### 3.1 Carbon yields of Rock-Eval® thermal analysis

Figure 1a presents the TOCre6 plotted against TOCea. We observed a high correlation despite a few points far from the regression line ($R^2=0.96$, n = 2037) and an average carbon yield, corresponding to the slope of the regression, equal to 86%. Limiting the Rock-Eval® dataset to samples passing our quality check regarding the Rock-Eval® organic carbon yields (yields ranging from 0.7 and 1.3) left out 145 samples. Another sample was left out because of its TOC content: with a value of 0.57 g kg-1, this sample contains too little organic carbon for the data from the Rock-Eval® thermal analysis to be routinely exploitable (Khedim et al., 2021). A principal components analysis (PCA) conducted on all the topsoil samples showed no cluster for the samples with poor organic carbon yields ("rejected") compared to the samples with good yields ("accepted") (Fig. D). However, there was a significant difference between the means of the two groups for many pedoclimatic parameters. In particular, the total sand content was on average 76% higher in the rejected samples compared to accepted samples (101% higher for the coarse sand and 35% higher for the fine sand) and the carbonate content was also 67% higher in the rejected samples compared to accepted samples.

The remaining sample selection logically showed a better agreement between TOCre6 and TOCea, yet with on average lower TOCre6 values compared with TOCea (Rock-Eval® organic carbon yield of 0.87, $R^2=0.99$, n = 1891; Fig. 1b). The inorganic carbon content for this sample selection was slightly overestimated by the Rock-Eval® thermal analysis (Rock-Eval® inorganic carbon yield of 1.07, $R^2=0.98$, n = 1891). Finally, the total carbon content measured with Rock-Eval® thermal analysis for this sample selection is consistent with the total carbon measured using the elemental analysis (Rock-Eval® total carbon yield of 0.96, $R^2=0.99$, n = 1891).

**Figure 1: Carbon yields of Rock-Eval® thermal analysis. (a) Organic carbon yield: TOCre6 as a function of TOCea for all analyzed RMQS topsoil (0–30 cm) samples; (b) Organic carbon yield: TOCre6 as a function of TOCea for the RMQS topsoil samples limited to those with Rock-Eval® organic carbon yields ranging from 0.7 to 1.3 ; (c) Inorganic carbon yield: MinC as a function of Cinorg for the RMQS topsoil samples limited to those with Rock-Eval® organic carbon yields ranging from 0.7 to 1.3 ;**





**and (d) Total carbon yield: TOCre6+Minc as a function of TOCea+Cinorg for the RMQS topsoil samples limited to those with Rock-Eval® organic carbon yields ranging from 0.7 to 1.3.**

## 3.2 Soil organic matter thermal stability in French topsoils and its relationships with land cover

The summary statistics of many different Rock-Eval® temperature parameters for the 1891 RMQS topsoil samples with satisfactory Rock-Eval® organic carbon yields are compiled in Supplementary Information (Table B).

Figure 2 shows the boxplots for the six selected parameters (T50_HC_PYR; T90_HC_PYR; T50_CO2_PYR; T50_CO2_OX; I-index; R-index) focusing on the four major land cover types.

**Figure 2: Effect of land cover on topsoil organic carbon thermal stability for the RMQS topsoil (0–30 cm) samples under the four major land covers in France: croplands, forests, grasslands, and vineyards & orchards. (a) T90_HC_PYR distribution; (b) T50_CO2_PYR distribution; (c) T50_CO2_OX distribution; (d) T50_HC_PYR distribution; (e) I-index distribution; (f) R-index**
**distribution. Samples are limited to those with Rock-Eval® organic carbon yields ranging from 0.7 to 1.3. For boxplots, the black midline of each box is the median. Lower and upper edges are respectively the 1st and 3rd quartiles. Lower and upper whiskers are respectively the maximum between the minimum value or the 1st quartile minus 1.5 times the interquartile range (max[min;Q1-1.5*(Q3-Q1)]) and the minimum between the maximum or the 3rd quartile plus 1.5 times the interquartile range (min[max;Q3+1.5*(Q3-Q1)]). Different letters indicate significant differences in the distribution of the values for the land uses**
**with the Kruskal-Wallis test (P < 0.05) and multiple Wilcoxon test (P < 0.05).**

We observed similar results for the temperature parameters T90_HC_PYR, T50_CO2_PYR and T50_CO2_OX: thermal stability was significantly higher in croplands and vineyards & orchards compared to forests and grasslands. Topsoil organic carbon was slightly but significantly less thermally stable in forests than in grasslands (Fig. 2a, 2b, 2c). Notably, three other Rock-Eval® parameters related to SOC thermal stability in the HC_PYR thermogram (T50_HC_PYR, I-index and R-index)
showed a different response to land cover (Fig. 2d, 2e, 2f). The T50_HC_PYR and R-index indicated no significant difference in thermal stability in forests and croplands. The I-index indicated a value significantly lower in forests than in croplands.

## 3.3 Elemental stoichiometry of soil organic matter in French topsoils and its relationships with land cover

The summary statistics of different elemental stoichiometry parameters for the 1891 RMQS topsoil samples with satisfactory
Rock-Eval® organic carbon yields are compiled in Supplementary Information (Table B). The HI (respectively OIre6 and C/N) mean value is 214 g HC.kg$^{-1}$ TOCre6 (respectively 177 g O$_2$.kg$^{-1}$ TOCre6 and 12.05).

We observed significantly higher average values for the HI in both grasslands and forests compared to croplands and vineyards & orchards (Fig. 3, Fig. Cb). In contrast, grasslands and forests showed smaller values of OIre6 compared to croplands and vineyards & orchards (Fig. 3, Fig. Cc).

In addition, Figure 3 highlights that the distribution of the C/N ratio on the Rock-Eval® pseudo van Krevelen diagrams (HI=f(OIre6)) depends on land cover. We observed a slight trend of the C/N ratio with the hydrogen and oxygen indices: the C/N ratio was higher for high HI and low OIre6. This trend was more pronounced for croplands and forests.

**Figure 3: Rock-Eval® pseudo van Krevelen diagrams (HI = f(OIre6)) for the RMQS topsoil (0–30 cm) samples in (a) croplands; (b) grasslands; (c) forests; and (d) vineyards & orchards. Colours indicate the values of the C/N ratio. Samples are limited to those**
**with Rock-Eval® organic carbon yields ranging from 0.7 to 1.3.**



## 3.4 Correlations between Rock-Eval® indicators of SOM thermal stability and elemental stoichiometry and pedoclimate

Table 1 presents the Spearman correlation coefficient values of the Rock-Eval® temperature and stoichiometric parameters with the selected pedoclimatic variables. The three selected temperature parameters (T90_HC_PYR, T50_CO2_PYR, T50_CO2_OX) correlated significantly and positively with the clay content and negatively with the sand content. T90_HC_PYR and T50_CO2_PYR also correlated positively with silt content, however with smaller correlation coefficient values. They strongly and positively (correlation coefficient > 0.3) correlated with the water pH, the carbonate content and the cation exchange capacity, while the relationships with iron oxyhydroxides content were much lower. The three selected temperature parameters were all significantly positively correlated to Mean Annual Temperature (MAT) and negatively correlated to Mean Annual Precipitation (MAP), although the correlations were weak.

**Table 1: Spearman correlation coefficients of the Rock-Eval® temperature and stoichiometric parameters with pedoclimatic variables (TOCre6, particle-size distribution, pH in water, carbonate content, cation exchange capacity (CEC), iron oxyhydroxides, mean annual temperature (MAT) averaged over 1969-1999 and mean annual precipitation (MAP) averaged over 1969-1999) for the RMQS topsoil (0–30 cm) samples. The analysis was limited to samples with Rock-Eval® organic carbon yields ranging from 0.7 to 1.3. Absolute values ≥ 0.3 are in bold. The asterisks indicate the p-value: 0 | *** | 0.001 | ** | 0.01 | * | 0.05 | ● | 0.1 | X.**

Regarding the indicators of SOM stoichiometry, HI and C/N correlated negatively with the clay and silt contents, contrary to OIre6 which correlated positively. HI and C/N also correlated negatively with the pH, the cation exchange capacity, and to a lesser extent with the carbonate content. They showed a slight negative correlation with the iron oxyhydroxides content measured by the Mehra-Jackson method. As for the thermal parameters, correlations with the climatic variables were on average smaller.

Additionally, the correlation coefficient of TOCre6 with HI was 0.35 (respectively -0.34 with OIre6, 0.37 with C/N, -0.26 with T90_HC_PYR, -0.21 with T50_CO2_PYR, -0.05 with T50_CO2_OX).

## 3.5 Distribution of some Rock-Eval® indicators of SOM thermal stability and elemental stoichiometry over the French mainland territory

Figure 4 shows the point maps of the HI and T50_CO2_PYR values over the French mainland territory. The missing topsoil samples (133 not included in the initial sample set and 146 rejected due to poor C yields) are distributed over the whole territory with some clusters in the north of the French Alps, north-east, Corsica, south-east and in the Landes. The first three clusters come from the 133 samples not included in the initial set. The Landes and south-east clusters are from both the absent samples and from the rejected samples: in particular, the soils in the Landes contain on average more sand, which is characteristic - as stated above - of the rejected samples. Visually, we noticed an autocorrelation of the values, HI and T50_CO2_PYR presenting on average opposite trends (the Spearman correlation coefficient between HI and T50_CO2_PYR is -0.69). Mountainous regions (notably the French Alps, the Pyrenees and the Massif Central) exhibit higher HI values and lower SOC thermal stability. Conversely, plain areas usually presented higher SOC thermal stability and lower HI values as in the Paris Basin and in the south-west and south-east part of the country. Brittany, Normandy and



the Landes are somewhat exceptions to this rule as they show high HI values and a relatively low SOC thermal stability.

Figure 5 shows the land cover at each sampling site.

**Figure 4: Point maps of two Rock-Eval® parameters: (a) Hydrogen Index values and (b) T50_CO2_PYR values on the French mainland territory for the RMQS topsoil (0–30 cm) samples, along with (c) the map of the main regions used for the**
**interpretation.**

**Figure 5: Map of the land cover at each sampling site. The number in brackets corresponds to the number of sites for each land cover in our final dataset (n=1891).**

## 4 Discussion

### 4.1 Carbon yields of Rock-Eval® thermal analysis

Our average organic carbon yield (0.86; Fig.1a) was in line with previous studies. Indeed, Disnar et al. (2003) obtained slightly higher yields (0.91), as well as Cécillon et al. (2018; 2021) (organic carbon yield from 0.90 to 0.96 depending on the sites) whereas Saenger et al. (2013) had lower yields (0.77). However, some samples presented high discrepancies between their TOCea and TOCre6 values. Samples with a TOCre6 value strongly differing from its corresponding TOCea value were systematically re-analyzed using Rock-Eval®, which confirmed their first TOCre6 measurement. The outliers regarding the
organic carbon yield were thus not related to a problem in their Rock-Eval® measurement. These very different values, which concern a few dozens of samples, could have different origins such as error on sample labeling, aliquoting, grinding or storage conditions. Indeed, for the same sample, the powders used for the elemental analysis and the Rock-Eval® thermal analysis did not come from the same aliquot. In addition, the elemental analyses were performed shortly after sampling, whereas the samples analyzed in Rock-Eval® were stored for about fifteen years. We can therefore expect slightly better
yields when elementary and Rock-Eval® analysis are performed with less time between both, and on the exact same powders. This is what we plan for the samples of the second RMQS sampling campaign. The very different values between TOCea and TOCre6 could also be due, for some samples, to a mismeasurement of the total carbonate content, leading to a miscalculation of the inorganic and organic carbon contents. This hypothesis could be plausible, as the mean value of the carbonate content was significantly higher in the rejected samples. The last hypothesis originates from the high content of
sand in the rejected samples: sandy samples are more heterogeneous, thus the material used to determine the TOCea is more likely to differ from the one used to determine the TOCre6, than when sand content is lower. Moreover, the physical state of organic matter in sandy soils can be different from other soils. Disnar et al. (2003) encountered "pellets" of SOM in sandy soils, which can strongly influence the results of TOCea and TOCre6.

The samples presenting a high discrepancy between TOCea and TOCre6 were not considered further in the analysis. As
stated above, we restricted our study to the samples with organic carbon yield ranging from 0.7 to 1.3. This subjective threshold is a quality threshold to ensure that the samples analyzed using Rock-Eval® were the same as the samples analyzed using elemental analysis on which rely all studies conducted on the first campaign of the RMQS. This selection only marginally improved the average organic carbon yield (0.87; Fig. 1b) and organic carbon was still underestimated by





Rock-Eval®. Conversely, inorganic carbon yield was slightly overestimated (1.07; Fig. 1c). As a result, the yield of total carbon (organic + inorganic carbon) was close to 1.00 (Fig. 1d). This suggests that almost all sample carbon is detected by the Rock-Eval® machine in the five thermograms but that a small part of the organic carbon is erroneously attributed to inorganic carbon. This may be due to a slight misplacement of the boundary between organic and inorganic carbon, probably in the S3 and S3CO signals. Also, the S3'CO signal is attributed half to organic carbon and half to inorganic carbon due to potential Boudouard reactions which is not always verified (Baudin et al., 2015; see e.g. Behar et al. (2001) for a definition

of the Rock-Eval® peaks). Of note, as MinC and TOCre6 are very well correlated to Cinorg and TOCea ($R^2 > 0.98$), it should therefore be possible to draw a correction formula to assess TOCea and Cinorg using Rock-Eval® with high accuracy. This would allow determining simultaneously in less than 1 h organic C and inorganic C with no risk of error due to erroneous decarbonation.

## 4.2 Thermal stability of soil organic carbon in French topsoils

We have observed that the thermal stability defined according to different Rock-Eval® parameters varies in French topsoils. We can investigate whether these variations are consistent with our knowledge of SOC biogeochemical stability. SOC biogeochemical stability is on average higher in croplands and vineyards compared to forest or grassland soils (Poeplau and Don, 2013). Indeed, fresh organic carbon inputs to soil are usually higher in forest and grassland compared to croplands where human exportation of biomass is higher (Murty et al., 2002). As a result, SOC fractions with lower mean residence

time in soils and lower thermal stability can be more abundant in forests and grasslands compared to croplands. For instance, several studies reported that carbon in particulate organic matter (a relatively more labile form of SOC) contributes more to total SOC in forest and grassland compared to cropland (e.g. Guo and Gifford, 2002; Poeplau et al., 2011; Poeplau and Don, 2013; Lugato et al. 2021). Moreover, agricultural practices may also speed up SOC mineralization further limiting the accumulation of labile SOC fractions. For instance, Balesdent et al. (1990) observed that the tillage practices lead to a

significantly higher mineralization than no tillage. Combining the effects of lower carbon inputs and mineralization-enhancing practices, croplands contain on average less biogeochemically labile SOC than forests and grasslands.

Thermal stability, as assessed using T90_HC_PYR, T50_CO2_PYR and T50_CO2_OX, was the highest in vineyards, orchards and croplands compared to forest and grassland soils (Fig. 2). These results suggest that, overall, SOC thermal stability as assessed using these Rock-Eval® parameters is related to SOC biogeochemical stability. This is in good

agreement with previous results obtained on smaller datasets (Barré et al., 2016; Poeplau et al., 2019; Cécillon et al., 2021). On the contrary, there was no consistent relationship between thermal stability and expected biogeochemical stability when the thermal stability was measured using T50_HC_PYR, R-index and I-index (Fig. 2). Cécillon et al. (2021) reported for soils with highly contrasted biogeochemical stability that the relationship between thermal stability and biogeochemical stability was weaker for T50_HC_PYR, R-index and I-index. Our results showed that this relationship even disappears when

considering data sets with more heterogeneous topsoil samples. The use of the Rock-Eval® temperature parameters





T90_HC_PYR, T50_CO2_PYR and T50_CO2_OX should therefore be preferred when seeking to measure thermal stability indicators directly related to biogeochemical stability.

T90_HC_PYR, T50_CO2_PYR and T50_CO2_OX were all strongly and positively correlated to clay content and negatively correlated to sand content (Table 1). In a previous study, Soucémarianadin et al. (2018) did not observe any correlation
between T50_CO2_OX and clay or sand content, however, their study was conducted on forest soils only and on a much reduced number of study sites. Soil clay fractions interact with microbial compounds which results in the formation of organo-mineral complexes which SOC has a high biogeochemical stability (e.g. Lehmann and Kleber, 2015). We can therefore hypothesize that clay-rich soils are also richer in biogeochemically stable carbon. The positive correlation between clay content and SOC thermal stability, and the good correlations between CEC, which depends on the first order of the clay
content, and SOC thermal stability would then be another illustration of the link between SOC thermal and biogeochemical stabilities. Iron oxides are mineral compounds that are also supposed to protect SOC from decomposition. To this respect, the inconsistent (Mehra-Jackson Iron) or even negative correlations (Tamm Iron) between T90_HC_PYR, T50_CO2_PYR and T50_CO2_OX and iron oxides were not expected. These weak correlations could be attributed to the fact that the range of iron oxides contents is relatively small in our set of topsoils.

T90_HC_PYR, T50_CO2_PYR and T50_CO2_OX were all positively correlated to pH. Such a correlation between T50_CO2_OX and pH was already observed by Soucémarianadin et al. (2018) for a set of French forest soils. Acidity may protect SOM from degradation by microorganisms (Clivot et al., 2021), by reducing their activity, which is actually observed in low pH acidic bogs. We can therefore hypothesize that acidity slows down SOM mineralization which can favor the accumulation of labile SOC components. As these labile SOC fractions would appear as thermally unstable, it would explain
the positive relationship between pH and Rock-Eval® indicators of SOC thermal stability.

T90_HC_PYR, T50_CO2_PYR and T50_CO2_OX showed weak but significant positive correlations with MAT averaged over 1969-1999 (Fig. 5). Such a correlation has also been observed in Soucémarianadin et al. (2018) for French forest soils. As soil microbial activity and thus SOC mineralization increase with temperature (Rey and Jarvis, 2006), we can expect SOC labile fractions to be more rapidly processed at higher temperature. It would be in line with the observed positive
correlations between MAT and the three selected thermal stability indicators. The relatively weak (Spearman rho value below 0.2) correlations can be due to the fact that MAT will also play on carbon inputs to the soil. Indeed, if higher SOC mineralization was balanced by increased biomass inputs it would mess up the relation between MAT and SOC biogeochemical stability. In a similar way, the weak negative correlation between MAP and thermal stability may be explained by the complex effect of MAP on SOC biogeochemical stability: increased soil moisture stimulates SOC
processing up to a certain point (Moyano et al., 2013) and influences net primary production and therefore the soil carbon inputs. In any case, the relationships between SOC and MAP or MAT are hard to disentangle (Chen et al., 2019). Another explanation for the weak values is that the climatic data was obtained on an 8km × 8km grid and do not have the same precision as if a weather station had been deployed at each site. This probably adds noise to the correlation.



The point map representing SOC thermal stability over mainland France (Fig. 4b) illustrates the relationships between SOC
thermal stability, land cover, climate and pedological variables. Mountainous regions (e.g. Massif Central, Alps, Pyrenees)
where forest, grassland, and low MAT dominate, and presented by Martin et al. (2011) as with relatively high SOC contents,
had a lower SOC thermal stability. Plains dominated by croplands with intensive agricultural practices and with relatively
low SOC contents such as the Paris Basin showed high SOC thermal stability. The southern part of France with warmer
MAT, dominated by vineyards and croplands, and relatively low SOC contents also presented high SOC thermal stability.
The lower SOC thermal stabilities observed in Brittany and Normandy (which are agricultural regions) could be explained
by the higher proportion of livestock. Therefore, in addition to the presence of grasslands in these regions, the cultivated
soils in Brittany and Normandy are more likely to receive repeated application of exogenous organic matter.

### 4.3 Elemental stoichiometry of soil organic matter in French topsoils

Higher values of HI and lower values of OIre6 were observed in forests and grasslands compared to croplands and
vineyards. This trend was observed in previous studies (Disnar et al., 2003; Saenger et al., 2013; Sebag et al., 2016). It also
confirms that HI and OIre6 can be good proxies of SOC biogeochemical stability. Indeed, as previously observed,
biogeochemically stable SOC is more oxidized and H-depleted (Barré et al., 2016; Poeplau et al., 2019; Cécillon et al.,
2021).

The pseudo van Krevelen diagrams (Fig. 3) showed a high variability of the C/N ratio between land cover classes: the C/N
ratio was higher in forest topsoils than in grasslands, as well as in croplands and vineyards. This is classically explained by
the fact that SOC is on average less processed in forests and grasslands compared to croplands and vineyards (Cotrufo et al.,
2019), as well as by the higher C/N ratio of the biomass inputs to soil in forests. Indeed, the biotransformation of organic
matter tends to lower its C/N ratio and oxidize it (Cleveland and Liptzin, 2007). This is in good agreement with the observed
trends of decreasing HI and increasing OIre6 with decreasing C/N (Fig. 3).

SOM elemental stoichiometry presented correlation patterns with land cover, climate and pedological variables that were
similar to those observed for SOM thermal stability. HI and OIre6 are respectively negatively and positively correlated to pH
(Table 1) as previously observed by Soucémarianadin et al. (2018) in French forest soils. This would be in line with acidity
slowing down the mineralization of H-enriched labile SOC fractions (Clivot et al., 2019). The negative correlation between
clay content and HI could be explained by the fact the presence of clays can promote the protection of microbially processed
H-depleted SOM. Similarly to what was observed for SOM thermal stability, relationships between elemental stoichiometry
and climate variables are weak, probably because climate plays on both soil carbon inputs and outputs in opposite ways
(climate conditions enhancing SOC mineralization usually also enhance fresh SOM inputs).

The point map of HI in mainland France (Fig. 4a) illustrated the effect of land cover, climate and pedological variables on
SOM elemental stoichiometry. Regions dominated by grassland and forest (Fig. 5) such as mountainous regions, the Landes
forest or the forest-dominated east part of France were characterized by a relatively H-enriched SOM. Conversely, regions
dominated by croplands, vineyards & orchards, and by high MAT were characterized by a relatively H-depleted SOM. Both

point maps of thermal stability and HI (Fig. 4) also illustrated the relationships previously observed between these Rock-Eval® parameters (Barré et al., 2016; Cécillon et al., 2021).

## 5 Conclusion

This study is an unprecedented effort to make widespread thermal analysis measurements on a national soil quality monitoring network. It demonstrated that Rock-Eval® may be used as a rapid and cost-effective method to assess the thermal stability and elemental stoichiometry of SOM on national soil monitoring networks. Our results highlighted the influence of land cover and pedoclimatic variables on SOM thermal stability and elemental stoichiometry. They suggested that some Rock-Eval® temperature parameters describing SOC thermal stability (T90_HC_PYR, T50_CO2_PYR and

T50_CO2_OX) could be used as reliable proxies of SOC biogeochemical stability, while others parameters could not (T50_HC_PYR, R-index and I-index). Our study also opened wide perspectives for future research. In the short term, these Rock-Eval® results on French topsoils can be used as input to the PARTYSOC machine-learning model (Cécillon et al. 2021) to infer the size of the centennially stable SOC fraction. They can also be compared to other proxies of SOC biogeochemical stability such as SOM physical fractionation results. In the medium term, it will be interesting to test

whether this analytical information can be used to improve the accuracy of SOC stock evolution simulations at the scale of a national soil monitoring network, as it was observed for the AMG model of SOC dynamics in several French long term agronomic experiments (Kanari et al., 2022).

## Data availability

Data on basic soil properties are freely available from the GIS Sol dataverse website:
https://data.inrae.fr/dataset.xhtml?persistentId=doi:10.15454/BNCXYB.

## Author contribution

Amicie Delahaie, Lauric Cécillon and Pierre Barré gathered the sample collection and ensured that samples were properly ground. Florence Savignac and François Baudin produced the Rock-Eval® thermal analyses. Dominique Arrouays, Antonio Bispo, Line Boulonne, Claudy Jolivet, Manuel Martin, Céline Ratié and Nicolas Saby provided the detailed pedoclimatic
data. Nicolas Saby produced the point maps. Amicie Delahaie processed and interpreted the data with the contribution of all co-authors. Amicie Delahaie, Pierre Barré and Lauric Cécillon wrote the manuscript with contribution of all the co-authors.

## Competing interests

The authors declare that they have no conflict of interest.



**Acknowledgements**

The École Normale Supérieure of Paris is greatly acknowledged for the funding of the PhD thesis grant of Amicie Delahaie.
The ADEME (Rock-Eval®-RMQS project, convention n°2003C0017) is acknowledged for their support.

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



|  | T90_HC_PYR | T50_CO2_PYR | T50_CO2_OX | HI | OIre6 | C/N |
|---|---|---|---|---|---|---|
| **TOCre6**<br>**(n=1891)** | -0.26<br>*** | -0.21<br>*** | -0.05<br>* | **0.35**<br>*** | **-0.34**<br>*** | **0.37**<br>*** |
| **clay**<br>**(n=1891)** | **0.46**<br>*** | **0.56**<br>*** | **0.45**<br>*** | **-0.35**<br>*** | **0.33**<br>*** | -0.27<br>*** |
| **silt (total)**<br>**(fine\|coarse)**<br>**(n=1891)** | 0.13<br>***<br>0.12 \| 0.09<br>*** \| *** | 0.20<br>***<br>0.29 \| 0.12<br>*** \| *** | -0.04<br>•<br>0.03 \| -0.09<br>x \| *** | -0.18<br>***<br>-0.16 \| -0.17<br>*** \| *** | 0.26<br>***<br>0.23 \| 0.23<br>*** \| *** | **-0.31**<br>***<br>-0.23 \| -0.28<br>*** \| *** |
| **sand (total)**<br>**(fine\|coarse)**<br>**(n=1891)** | **-0.36**<br>***<br>-0.17 \| -0.34<br>*** \| *** | **-0.42**<br>*** | -0.25<br>*** | 0.31<br>*** | -0.34<br>*** | 0.35<br>*** |
| **water pH**<br>**(n=1891)** | **0.71**<br>*** | **0.73**<br>*** | **0.44**<br>*** | **-0.42**<br>*** | **0.39**<br>*** | **-0.52**<br>*** |
| **carbonates**<br>**(n=1891)** | **0.53**<br>*** | **0.56**<br>*** | **0.45**<br>*** | -0.20<br>*** | 0.24<br>*** | -0.28<br>*** |
| **CEC**<br>**(n=1891)** | **0.60**<br>*** | **0.56**<br>*** | **0.46**<br>*** | **-0.36**<br>*** | **0.32**<br>*** | **-0.36**<br>*** |
| **Free iron (Tamm)**<br>**(n=1622)** | -0.16<br>*** | -0.13<br>*** | -0.26<br>*** | -0.06<br>* | 0.09<br>*** | -0.07<br>** |
| **Free iron (Mehra-Jackson)**<br>**(n=1621)** | 0.08<br>** | **0.36**<br>*** | -0.05<br>* | **-0.33**<br>*** | **0.35**<br>*** | -0.11<br>*** |
| **MAT (1969-1999)**<br>**(n=1891)** | 0.12<br>*** | 0.24<br>*** | 0.10<br>*** | -0.20<br>*** | 0.06<br>** | -0.11<br>*** |
| **MAP (1969-1999)**<br>**(n=1891)** | -0.25<br>*** | -0.09<br>*** | -0.20<br>*** | 0.14<br>*** | -0.10<br>*** | 0.21<br>*** |

**Table 1: Spearman correlation coefficients of the Rock-Eval® temperature and stoichiometric parameters with pedoclimatic variables (TOCre6, particle-size distribution, pH in water, carbonate content, cation exchange capacity (CEC), iron oxyhydroxides, mean annual temperature (MAT) averaged over 1969-1999 and mean annual precipitation (MAP) averaged over 1969-1999) for the RMQS topsoil (0–30 cm) samples. The analysis was limited to samples with Rock-Eval® organic carbon yields ranging from 0.7 to 1.3. Absolute values ≥ 0.3 are in bold. The asterisks indicate the p-value: 0 | *** | 0.001 | ** | 0.01 | * | 0.05 | ● |**
**0.1 | X.**





**Figure 1: Carbon yields of Rock-Eval® thermal analysis. (a) Organic carbon yield: TOCre6 as a function of TOCea for all analyzed RMQS topsoil (0–30 cm) samples; (b) Organic carbon yield: TOCre6 as a function of TOCea for the RMQS topsoil samples limited to those with Rock-Eval® organic carbon yields ranging from 0.7 to 1.3 ; (c) Inorganic carbon yield: MinC as a function of Cinorg for the RMQS topsoil samples limited to those with Rock-Eval® organic carbon yields ranging from 0.7 to 1.3 ; and (d) Total carbon yield: TOCre6+Minc as a function of TOCea+Cinorg for the RMQS topsoil samples limited to those with Rock-Eval® organic carbon yields ranging from 0.7 to 1.3.**





**Figure 2: Effect of land cover on topsoil organic carbon thermal stability for the RMQS topsoil (0–30 cm) samples under the four major land covers in France: croplands, forests, grasslands, and vineyards & orchards. (a) T90_HC_PYR distribution; (b) T50_CO2_PYR distribution; (c) T50_CO2_OX distribution; (d) T50_HC_PYR distribution; (e) I-index distribution; (f) R-index distribution. Samples are limited to those with Rock-Eval® organic carbon yields ranging from 0.7 to 1.3. For boxplots, the black midline of each box is the median. Lower and upper edges are respectively the 1st and 3rd quartiles. Lower and upper whiskers are respectively the maximum between the minimum value or the 1st quartile minus 1.5 times the interquartile range (max[min;Q1-1.5*(Q3-Q1)]) and the minimum between the maximum or the 3rd quartile plus 1.5 times the interquartile range (min[max;Q3+1.5*(Q3-Q1)]). Different letters indicate significant differences in the distribution of the values for the land uses with the Kruskal-Wallis test (P < 0.05) and multiple Wilcoxon test (P < 0.05).**



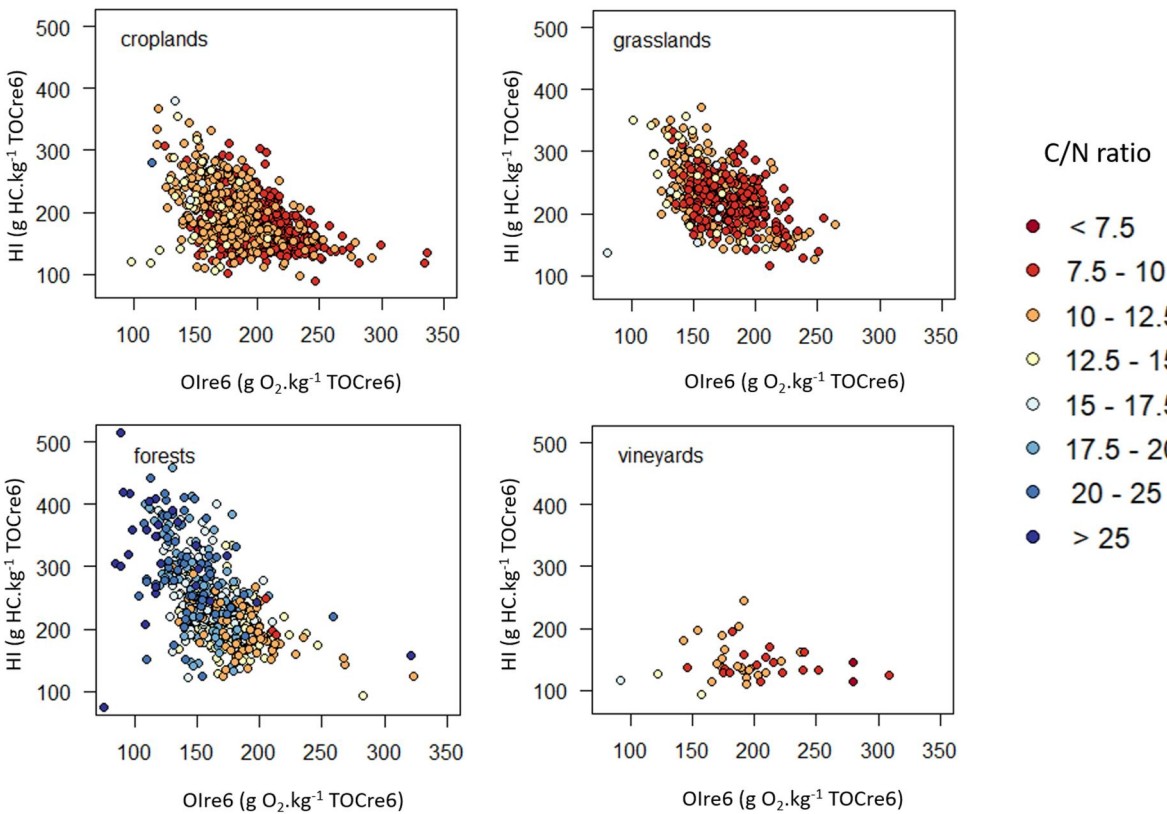

**Figure 3: Rock-Eval® pseudo van Krevelen diagrams (HI = f(OIre6)) for the RMQS topsoil (0–30 cm) samples in (a) croplands; (b) grasslands; (c) forests; and (d) vineyards & orchards. Colours indicate the values of the C/N ratio. Samples are limited to those with Rock-Eval® organic carbon yields ranging from 0.7 to 1.3.**





**Figure 4: Point maps of two Rock-Eval® parameters: (a) Hydrogen Index values and (b) T50_CO2_PYR values on the French mainland territory for the RMQS topsoil (0–30 cm) samples, along with (c) the map of the main regions used for interpretation.**

685



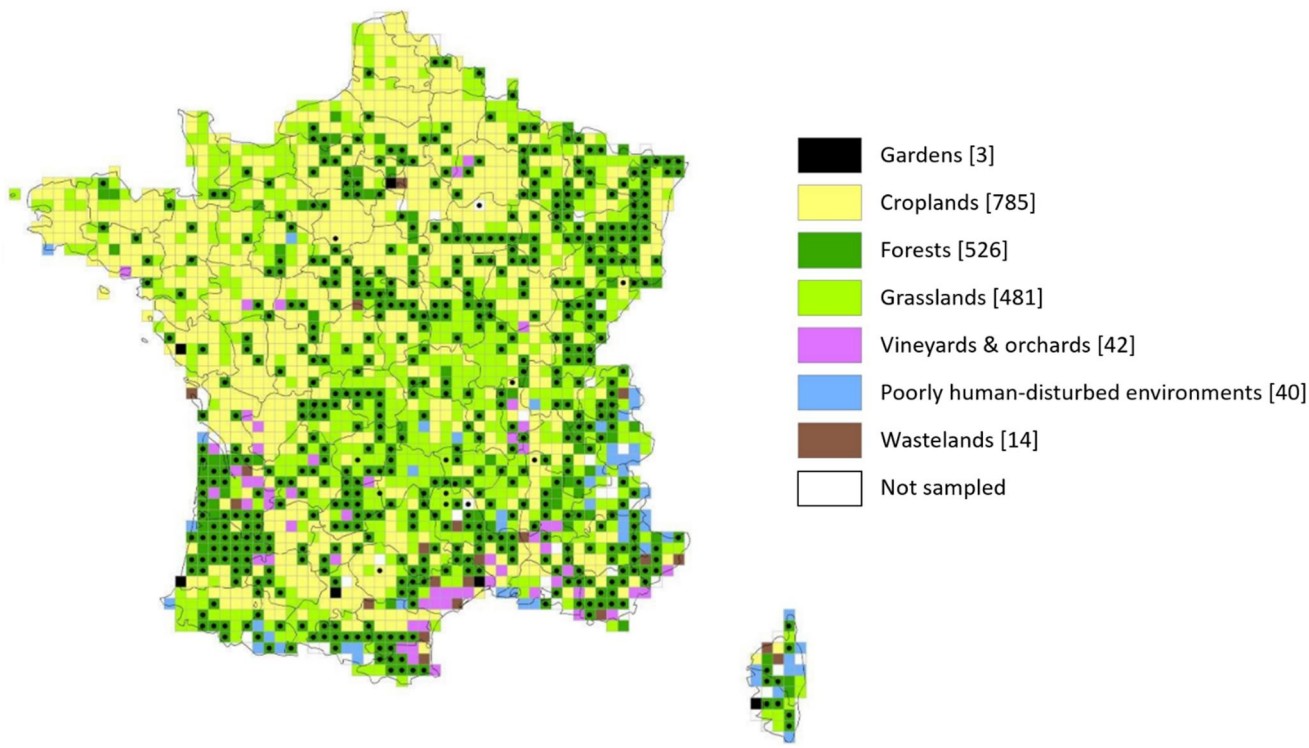

**Figure 5: Map of the land cover at each sampling site (Jolivet, 2011). The number in brackets corresponds to the number of sites for each land cover in our final dataset (n=1891).**





690 **Appendices**

| parameter | unit | formula | description |
|---|---|---|---|
| TOCre6 | g.kg$^{-1}$ sample | $PC \times 10 + S4CO \times \frac{12}{28}$ | total organic carbon |
| MinC | g.kg$^{-1}$ sample | $S3' \times \frac{12}{44} + \frac{S3'CO}{2} \times \frac{12}{28} + S5 \times \frac{12}{44}$ | total inorganic carbon |
| PC | g.kg$^{-1}$ sample | $\frac{(S1+S2) \times 0.83 + S3 \frac{12}{44} + (S3CO + \frac{S3'CO}{2}) \times \frac{12}{28}}{10}$ | amount of pyrolyzable organic carbon |
| PC/TOCre6 | no unit | $\frac{PC}{TOCre6}$ | ratio of pyrolyzable organic carbon over total organic carbon |
| PseudoS1 | g.kg$^{-1}$ sample | Integration of the thermogram (see Behar et al., 2001 for thermograms descriptions) | carbon released during the first pyrolysis isotherm |
| S2 | g.kg$^{-1}$ sample | Integration of the thermogram (see Behar et al., 2001 for thermograms descriptions) | carbon released as hydrocarbons during the pyrolysis except during the first isotherm |
| S2/PC | no unit | $\frac{S2}{PC}$ | ratio of carbon released as hydrocarbons during the pyrolysis except during the first isotherm over the pyrolyzable organic carbon |
| T50_HC_PYR | °C | Integration of the thermogram (see Behar et al., 2001 for thermograms descriptions) to obtain temperature | temperature at which 50 % of the hydrocarbon effluents have been emitted during the pyrolysis ramp (the initial isotherm is excluded; the integration ends at 650°C) |
| T70_HC_PYR | °C | Integration of the thermogram (see Behar et al., 2001 for thermograms descriptions) to obtain temperature | temperature at which 70 % of the hydrocarbon effluents have been emitted during the pyrolysis ramp (the initial isotherm is excluded; the integration ends at 650°C) |
| T90_HC_PYR | °C | Integration of the thermogram (see Behar et al., 2001 for thermograms descriptions) to obtain temperature | temperature at which 90 % of the hydrocarbon effluents have been emitted during the pyrolysis ramp (the initial isotherm is excluded; the integration ends at 650°C) |
| T30_CO2_PYR | °C | Integration of the thermogram (see Behar et al., 2001 for thermograms descriptions) to obtain temperature | temperature at which 30 % of the CO2 have been emitted during the pyrolysis ramp (the beginning isotherm is excluded; the integration ends at 560° C) |
| T50_CO2_PYR | °C | Integration of the thermogram (see Behar et al., 2001 for thermograms descriptions) to obtain temperature | temperature at which 50 % of the CO2 have been emitted during the pyrolysis ramp (the beginning isotherm is excluded; the integration ends at 560° C) |
| T70_CO2_PYR | °C | Integration of the thermogram (see Behar et al., 2001 for thermograms descriptions) to obtain temperature | temperature at which 70 % of the CO2 have been emitted during the pyrolysis ramp (the beginning isotherm is excluded; the integration ends at 560° C) |
| T90_CO2_PYR | °C | Integration of the thermogram (see Behar et al., 2001 for thermograms descriptions) to obtain temperature | temperature at which 90 % of the CO2 have been emitted during the pyrolysis ramp (the beginning isotherm is excluded; the integration ends at 560° C) |
| T50_CO_PYR | °C | Integration of the thermogram (see Behar et al., 2001 for thermograms descriptions) to obtain temperature | temperature at which 50 % of the CO have been emitted during the pyrolysis ramp (the beginning isotherm is excluded; the integration ends at 560°C) |
| T50_CO2_OX | °C | Integration of the thermogram (see Behar et al., 2001 for thermograms descriptions) to obtain temperature | temperature at which 50 % of the CO2 have been emitted during the oxidation phase (the integration ends at 611°C) |
| T70_CO2_OX | °C | Integration of the thermogram (see Behar et al., 2001 for thermograms descriptions) to obtain temperature | temperature at which 70 % of the CO2 have been emitted during the oxidation phase (the integration ends at 611°C) |
| T90_CO2_OX | °C | Integration of the thermogram (see Behar et al., 2001 for thermograms descriptions) to obtain temperature | temperature at which 90 % of the CO2 have been emitted during the oxidation phase (the integration ends at 611°C) |
| T50_CO_OX | °C | Integration of the thermogram (see Behar et al., 2001 for thermograms descriptions) to obtain temperature | temperature at which 50 % of the CO have been emitted during the oxidation phase (the integration ends at 850°C) |
| T70_CO_OX | °C | Integration of the thermogram (see Behar et al., 2001 for thermograms descriptions) to obtain temperature | temperature at which 70 % of the CO have been emitted during the oxidation phase (the integration ends at 850°C) |
| I-index | no unit | Integration of the thermogram (see Sebag et al., 2016 for boundaries) | related to the thermolabile organic carbon released as hydrocarbon effluents, see Sebag et al., 2016 |
| R-index | no unit | Integration of the thermogram (see Sebag et al., 2016 for boundaries) | proportion of thermostable organic carbon released as hydrocarbon effluents after 400°C, see Sebag et al., 2016 |
| HI | g HC.kg$^{-1}$ TOCre6 | $\frac{S2 \times 100}{TOCre6}$ | ratio of emitted hydrocarbons to TOCre6 |
| OIre6 | g O$_2$.kg$^{-1}$ TOCre6 | $\frac{16}{28} \times \frac{S3CO \times 100}{TOCre6} + \frac{32}{44} \times \frac{S3 \times 100}{TOCre6}$ | ratio of organic oxygen to TOCre6 |

**Table A: Description of the Rock-Eval® parameters and their calculation.**





| | min | | | | | mean | | | | | max | | | | |
|---|---|---|---|---|---|---|---|---|---|---|---|---|---|---|---|
| | total | croplands | grasslands | forests | vineyards & orchards | total | croplands | grasslands | forests | vineyards & orchards | total | croplands | grasslands | forests | vineyards & orchards |
| C_N | 6,9 | 7,0 | 8,2 | 8,9 | 6,9 | 12,1 | 10,1 | 10,5 | 16,7 | 10,5 | 50,8 | 22,6 | 17,4 | 50,8 | 17,4 |
| T50_HC_PYR | 385 | 403 | 385 | 392 | 403 | 424 | 426 | 419 | 425 | 432 | 466 | 466 | 450 | 459 | 465 |
| T70_HC_PYR | 438 | 445 | 438 | 438 | 453 | 464 | 466 | 461 | 463 | 473 | 508 | 508 | 488 | 492 | 506 |
| T90_HC_PYR | 483 | 494 | 490 | 483 | 504 | 515 | 521 | 511 | 508 | 534 | 572 | 567 | 552 | 549 | 570 |
| T30_CO2_PYR | 311 | 325 | 318 | 311 | 327 | 337 | 341 | 335 | 331 | 348 | 448 | 381 | 385 | 350 | 379 |
| T50_CO2_PYR | 347 | 363 | 359 | 347 | 372 | 384 | 390 | 382 | 377 | 399 | 497 | 447 | 458 | 403 | 441 |
| T70_CO2_PYR | 389 | 411 | 408 | 389 | 423 | 437 | 442 | 435 | 429 | 453 | 527 | 500 | 509 | 461 | 496 |
| T90_CO2_PYR | 462 | 485 | 484 | 462 | 495 | 505 | 509 | 505 | 499 | 517 | 550 | 542 | 544 | 529 | 542 |
| T50_CO_PYR | 199 | 199 | 338 | 377 | 386 | 404 | 402 | 401 | 408 | 404 | 472 | 472 | 470 | 464 | 427 |
| T50_CO_OX | 326 | 350 | 326 | 329 | 370 | 401 | 410 | 393 | 391 | 433 | 586 | 586 | 529 | 492 | 545 |
| T70_CO_OX | 356 | 386 | 363 | 356 | 408 | 457 | 473 | 448 | 439 | 504 | 690 | 666 | 690 | 551 | 611 |
| T50_CO2_OX | 377 | 388 | 377 | 377 | 393 | 413 | 416 | 409 | 410 | 425 | 493 | 480 | 485 | 467 | 463 |
| T70_CO2_OX | 404 | 417 | 406 | 404 | 436 | 461 | 468 | 455 | 454 | 484 | 554 | 539 | 534 | 512 | 528 |
| T90_CO2_OX | 444 | 465 | 464 | 444 | 518 | 530 | 537 | 528 | 521 | 548 | 597 | 585 | 577 | 563 | 577 |
| PseudoS1 | 0,02 | 0,02 | 0,03 | 0,04 | 0,02 | 0,14 | 0,09 | 0,15 | 0,18 | 0,08 | 0,98 | 0,33 | 0,64 | 0,98 | 0,22 |
| S2 | 0,14 | 0,59 | 0,85 | 0,14 | 0,34 | 4,22 | 2,30 | 5,12 | 5,95 | 1,18 | 61,6 | 8,26 | 27,3 | 32,9 | 5,53 |
| S2_PC | 0,31 | 0,31 | 0,46 | 0,38 | 0,40 | 0,65 | 0,62 | 0,68 | 0,69 | 0,54 | 0,9 | 0,82 | 0,84 | 0,9 | 0,68 |
| HI | 67 | 88 | 116 | 74 | 94 | 214 | 190 | 229 | 240 | 144 | 515 | 379 | 372 | 515 | 244 |
| OIre6 | 75 | 98 | 80 | 75 | 92 | 177 | 189 | 171 | 163 | 198 | 337 | 337 | 264 | 323 | 325 |
| HI_OIre6 | 0,32 | 0,36 | 0,51 | 0,33 | 0,39 | 1,28 | 1,05 | 1,39 | 1,57 | 0,76 | 5,82 | 3,06 | 3,46 | 5,82 | 1,27 |
| PC | 0,33 | 0,95 | 1,56 | 0,33 | 0,77 | 6,14 | 3,62 | 7,32 | 8,45 | 2,12 | 73,5 | 11,9 | 36,3 | 43,6 | 8,09 |
| PC_TOCre6 | 0,1 | 0,14 | 0,16 | 0,1 | 0,15 | 0,27 | 0,25 | 0,28 | 0,28 | 0,22 | 0,48 | 0,4 | 0,38 | 0,48 | 0,3 |
| TOCre6 | 1,1 | 3,6 | 5,4 | 1,1 | 2,8 | 22,5 | 14,1 | 26,3 | 30,2 | 9,5 | 213 | 46,5 | 127 | 144 | 27,3 |
| MINC | 0,2 | 0,2 | 0,3 | 0,2 | 0,5 | 8,0 | 9,7 | 6,4 | 6,4 | 12,4 | 108 | 108 | 85,8 | 97,9 | 58,3 |
| R_index | 0,44 | 0,51 | 0,44 | 0,47 | 0,51 | 0,61 | 0,62 | 0,58 | 0,61 | 0,65 | 0,77 | 0,77 | 0,71 | 0,77 | 0,77 |
| I_index | -0,14 | -0,06 | -0,01 | -0,14 | -0,06 | 0,13 | 0,12 | 0,18 | 0,11 | 0,08 | 0,39 | 0,31 | 0,39 | 0,36 | 0,32 |





| | First quartile | | | | | Median | | | | | Third quartile | | | | | Standard deviation | | | | |
|---|---|---|---|---|---|---|---|---|---|---|---|---|---|---|---|---|---|---|---|---|
| | total | croplands | grasslands | forests | vineyards & orchards | total | croplands | grasslands | forests | vineyards & orchards | total | croplands | grasslands | forests | vineyards & orchards | total | croplands | grasslands | forests | vineyards & orchards |
| C_N | 9,7 | 9,4 | 9,7 | 13,6 | 9,4 | 10,5 | 9,8 | 10,2 | 15,5 | 10,6 | 13,2 | 10,4 | 10,9 | 18,3 | 11,2 | 4,00 | 1,21 | 1,18 | 4,86 | 1,78 |
| T50_HC_PYR | 418 | 420 | 413 | 420 | 425 | 424 | 425 | 419 | 427 | 432 | 431 | 433 | 425 | 432 | 440 | 10,3 | 9,5 | 10,1 | 9,4 | 11,3 |
| T70_HC_PYR | 459 | 461 | 456 | 459 | 465 | 463 | 465 | 460 | 463 | 473 | 468 | 471 | 465 | 467 | 481 | 8,4 | 8,7 | 7,3 | 7,0 | 11,4 |
| T90_HC_PYR | 506 | 512 | 505 | 500 | 523 | 513 | 520 | 510 | 508 | 534 | 523 | 529 | 516 | 515 | 542 | 12,7 | 11,9 | 9,1 | 10,5 | 13,9 |
| T30_CO2_PYR | 332 | 336 | 331 | 327 | 341 | 337 | 340 | 335 | 332 | 346 | 341 | 345 | 338 | 337 | 353 | 9,0 | 7,0 | 6,8 | 7,2 | 11,0 |
| T50_CO2_PYR | 377 | 383 | 376 | 371 | 389 | 384 | 389 | 381 | 378 | 396 | 390 | 395 | 387 | 384 | 406 | 11,9 | 9,7 | 9,6 | 9,9 | 15,4 |
| T70_CO2_PYR | 429 | 435 | 428 | 421 | 441 | 436 | 441 | 434 | 429 | 451 | 443 | 448 | 440 | 437 | 463 | 13,3 | 11,3 | 10,4 | 11,9 | 16,9 |
| T90_CO2_PYR | 499 | 503 | 500 | 493 | 509 | 505 | 508 | 504 | 500 | 517 | 511 | 514 | 509 | 507 | 525 | 10,3 | 8,5 | 7,7 | 10,7 | 11,1 |
| T50_CO_PYR | 399 | 400 | 398 | 402 | 400 | 403 | 402 | 401 | 406 | 404 | 406 | 405 | 403 | 412 | 408 | 11,8 | 11,4 | 7,8 | 11,6 | 7,0 |
| T50_CO_OX | 382 | 388 | 377 | 376 | 403 | 397 | 405 | 391 | 394 | 427 | 413 | 424 | 407 | 405 | 456 | 29,4 | 30,8 | 24,8 | 22,6 | 39,1 |
| T70_CO_OX | 426 | 446 | 421 | 418 | 482 | 453 | 475 | 444 | 435 | 506 | 486 | 499 | 475 | 454 | 523 | 39,9 | 37,8 | 36,5 | 32,4 | 40,6 |
| T50_CO2_OX | 403 | 405 | 402 | 403 | 412 | 411 | 413 | 408 | 411 | 427 | 420 | 425 | 416 | 417 | 433 | 13,7 | 14,5 | 11,9 | 11,1 | 18,0 |
| T70_CO2_OX | 444 | 449 | 443 | 439 | 459 | 457 | 468 | 453 | 451 | 491 | 478 | 486 | 466 | 466 | 501 | 22,7 | 22,8 | 18,0 | 21,0 | 25,9 |
| T90_CO2_OX | 518 | 528 | 519 | 503 | 538 | 534 | 540 | 531 | 524 | 551 | 544 | 548 | 539 | 540 | 554 | 20,7 | 17,5 | 16,9 | 23,0 | 13,9 |
| PseudoS1 | 0,08 | 0,06 | 0,10 | 0,11 | 0,06 | 0,11 | 0,08 | 0,13 | 0,15 | 0,08 | 0,16 | 0,11 | 0,17 | 0,22 | 0,10 | 0,09 | 0,04 | 0,08 | 0,11 | 0,04 |
| S2 | 1,72 | 1,27 | 2,73 | 2,92 | 0,69 | 2,99 | 1,82 | 3,96 | 4,47 | 0,94 | 5,05 | 2,88 | 6,00 | 7,53 | 1,46 | 4,310 | 1,44 | 3,96 | 4,56 | 0,880 |
| S2_PC | 0,6 | 0,57 | 0,64 | 0,64 | 0,50 | 0,66 | 0,62 | 0,69 | 0,69 | 0,53 | 0,71 | 0,66 | 0,73 | 0,75 | 0,58 | 0,08 | 0,07 | 0,06 | 0,08 | 0,07 |
| HI | 174 | 159 | 199 | 192 | 127 | 206 | 183 | 226 | 229 | 137 | 246 | 212 | 257 | 275 | 155 | 57,0 | 43,5 | 45,0 | 66,1 | 28,9 |
| OIre6 | 155 | 169 | 151 | 145 | 175 | 175 | 187 | 168 | 163 | 192 | 195 | 207 | 189 | 181 | 213 | 31,5 | 29,6 | 26,6 | 29,7 | 44,8 |
| HI_OIre6 | 0,91 | 0,82 | 1,09 | 1,09 | 0,62 | 1,17 | 0,98 | 1,34 | 1,39 | 0,69 | 1,52 | 1,21 | 1,63 | 1,83 | 0,86 | 0,55 | 0,37 | 0,43 | 0,69 | 0,23 |
| PC | 2,90 | 2,22 | 4,17 | 4,26 | 1,43 | 4,54 | 3,04 | 5,97 | 6,53 | 1,99 | 7,22 | 4,49 | 8,63 | 10,80 | 2,47 | 5,59 | 1,94 | 5,14 | 6,09 | 1,26 |
| PC_TOCre6 | 0,24 | 0,23 | 0,25 | 0,25 | 0,21 | 0,26 | 0,25 | 0,27 | 0,28 | 0,22 | 0,29 | 0,27 | 0,30 | 0,31 | 0,24 | 0,04 | 0,03 | 0,03 | 0,050 | 0,029 |
| TOCre6 | 11,2 | 9,1 | 15,3 | 15,4 | 6,7 | 17,0 | 12,5 | 21,8 | 24,3 | 8,5 | 26,8 | 17,1 | 30,3 | 38,4 | 11,4 | 18,5 | 7,05 | 17,5 | 20,9 | 4,70 |
| MINC | 1,0 | 0,9 | 1,1 | 1,0 | 0,8 | 1,6 | 1,3 | 1,8 | 1,7 | 2,1 | 4,3 | 5,2 | 3,4 | 3,7 | 15,9 | 16,7 | 19,83 | 13 | 14 | 18,1 |
| R_index | 0,57 | 0,58 | 0,55 | 0,58 | 0,62 | 0,61 | 0,61 | 0,58 | 0,62 | 0,65 | 0,64 | 0,65 | 0,61 | 0,65 | 0,68 | 0,05 | 0,04 | 0,05 | 0,05 | 0,05 |
| I_index | 0,08 | 0,08 | 0,13 | 0,05 | 0,01 | 0,13 | 0,13 | 0,19 | 0,11 | 0,09 | 0,19 | 0,17 | 0,23 | 0,16 | 0,12 | 0,08 | 0,06 | 0,07 | 0,08 | 0,07 |

695  **Table B: (2 pages) Minimum, maximum, mean, 1st quartile, 3rd quartile, median and standard deviation values of the Rock-Eval® parameters for the RMQS topsoil (0–30 cm) samples limited to those with Rock-Eval® organic carbon yields ranging from 0.7 to 1.3 (n=1891, "total") and by land cover.**





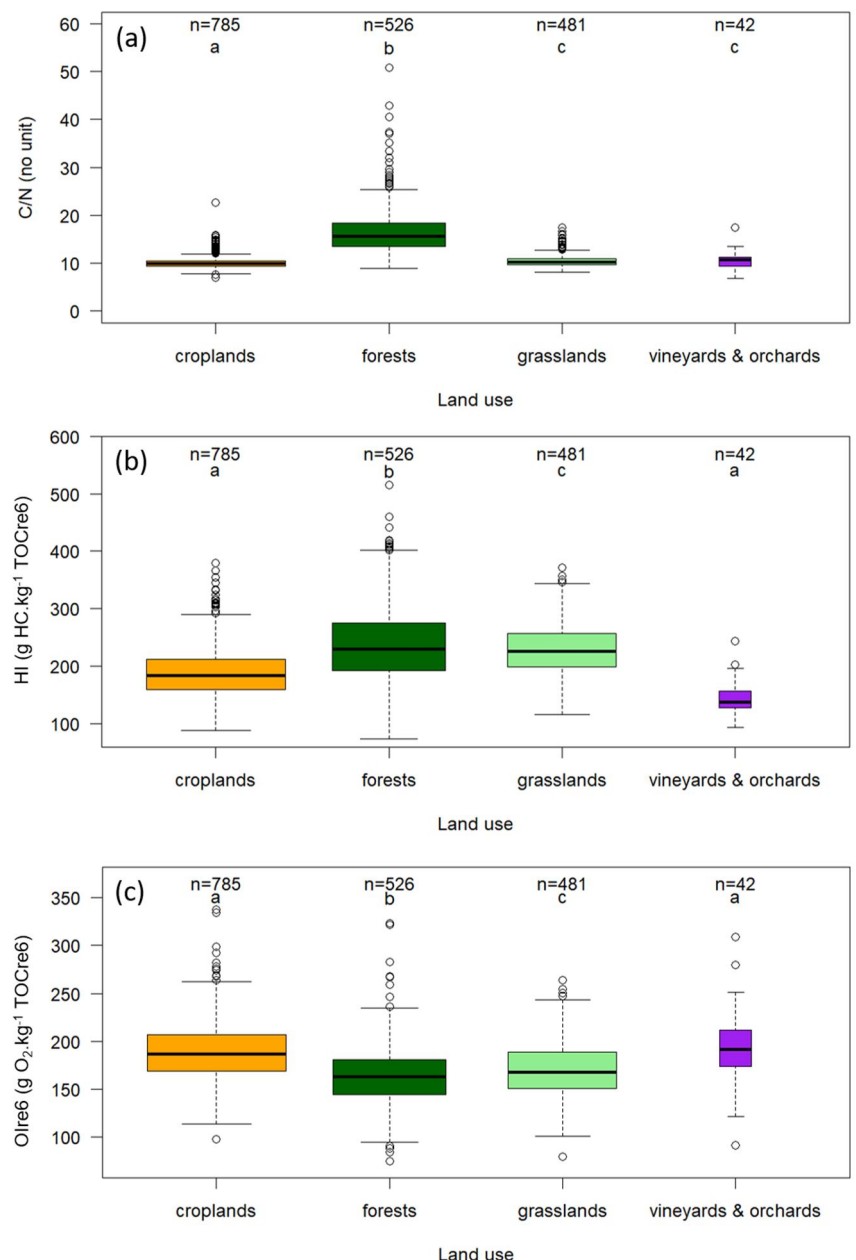

**Figure C: Effect of land cover on topsoil organic carbon stoichiometry for the RMQS topsoil (0–30 cm) samples under the four major land covers in France: croplands, forests, grasslands, and vineyards & orchards. (a) C/N distribution; (b) HI distribution; (c) OIre6 distribution. Samples are limited to those with Rock-Eval® organic carbon yields ranging from 0.7 to 1.3. For boxplots, the black midline of each box is the median. Lower and upper edges are respectively the 1st and 3rd quartiles. Lower and upper whiskers are respectively the maximum between the minimum value or the 1st quartile minus 1.5 times the interquartile range (max[min;Q1-1.5*(Q3-Q1)]) and the minimum between the maximum or the 3rd quartile plus 1.5 times the interquartile range (min[max;Q3+1.5*(Q3-Q1)]). Different letters indicate significant differences in the distribution of the values for the land uses with the Kruskal-Wallis test (P < 0.05) and multiple Wilcoxon test (P < 0.05).**





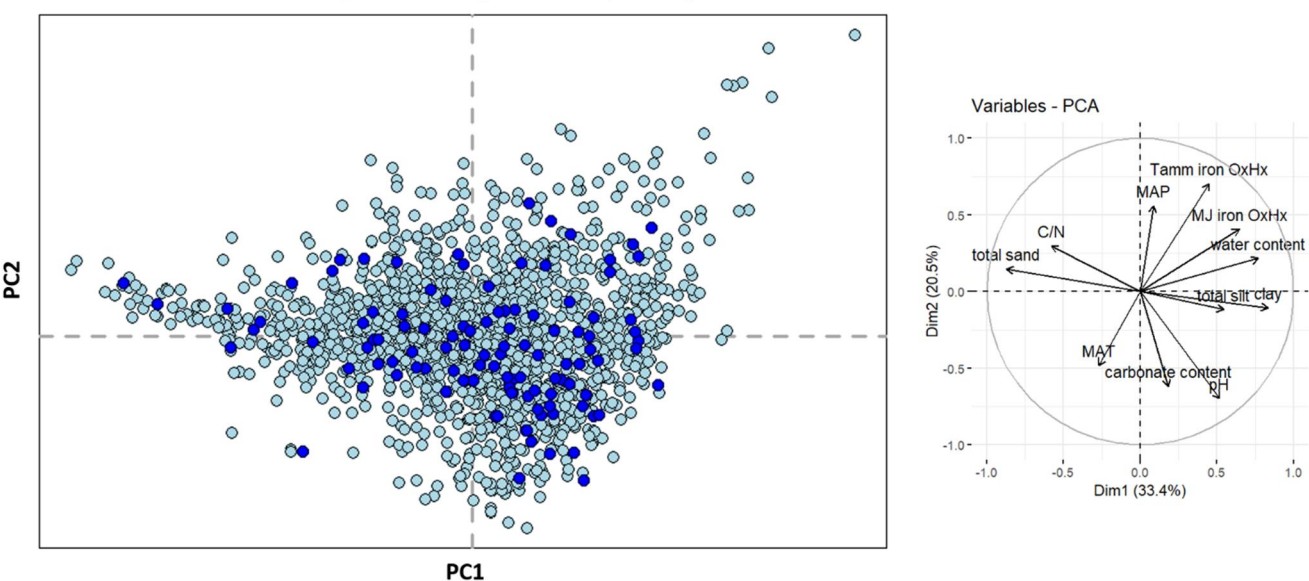

Figure D: Score of the 2037 samples on axes 1 and 2 of the principal component analysis on 11 pedoclimatic parameters: clay, total silt and total sand contents, pH in water, water content, carbonate content, mean annual temperature, mean annual precipitation, Tamm and Mehra-Jackson iron oxyhydroxides contents, C/N ratio for the RMQS topsoil (0–30 cm) samples. The samples with an organic carbon yield between 0.7 and 1.3 are plotted in light blue. The samples with an organic carbon yield < 0.7 or > 1.3 are plotted in dark blue.