# Peer review of "Elemental stoichiometry and Rock-Eval® thermal stability of organic matter in French topsoils"

_EGUsphere, 2022_

## Author Response (AR1)

**Point-by-point responses to Topical Editor's and R2's comments**

**Reply on RC2**

We thank the reviewer for evaluating our manuscript and for their constructive comments. Please find below our answers (plain text) to each specific comment (italics). The modifications effectively applied to the manuscript appear in color.

*I found this to be a very useful paper highlighting the utility of Rock-eval pyrolysis across a wide geography. I must admit that I was not surprised by any of the results. This was reassuring because it does show that cropland carbon tends to be more stabilised than forest or grassland carbon. I did appreciate the transparent approach to sample curation and that certain samples were removed. I also appreciate the questioning of the partition between organic and inorganic carbon, particularly with respect to the way the S3 curve is segmented. I think this is remains an ongoing question, and does, in part, contribute to the underestimation of organic carbon using this method. I did also appreciate the map of France build with this data, and I suggest below to combine Figures to more easily link land use and Rock eval parameters.*

*What follows are items that I feel ought to be addressed to improve this work:*

*For the land cover designations, only crop, grass, forest, and vineyards were addressed in detail. How were wastelands, human-disturbed and gardens grouped? Were they omitted from all analyses?*

**Answer:** All the samples available, regardless of their land cover, were analyzed using Rock-Eval thermal analysis. We analyzed 40 samples coming from poorly human disturbed sites, 14 from wastelands and 3 from gardens. Considering the very small number of samples for wastelands and gardens compared to the whole set, we decided not to include them in our statistical treatment. The number of poorly human-disturbed samples can be considered sufficient for statistical treatment, however they represent a very heterogeneous set of samples (10 miscellaneous subclasses such as peatlands, alpine grasslands, water edge vegetation, heath, dry siliceous meadows, etc.). We did not consider it relevant to analyze them as a whole. We will add a few lines to explain our choice in the revised version.

**Edit:** We explained our choice in the revised version to make it clearer why these land covers were not considered in the statistical treatment.

*Table 1, would like to also see a column for T50_HC_PYR*

**Answer:** The column for T50_HC_PYR will be added in the revised version of the manuscript.

**Edit:** The column for T50_HC_PYR has been added.

*Figure 2, perhaps I missed, are the box widths proportional to n?*

**Answer:** The box width is proportional to the square root of n; we will specify this in the caption in the revised version.

**Edit:** We added the information in the caption of Fig.2 and Fig. C.

*Figure 2, letters indicating significance appear incorrect. For example, Fig 2 (a), if they are all significantly different, should read cabd in that order. I do have doubts that forests and grasslands in that panel are significantly different. Please check the other panels as well.*

**Answer:** Indeed, we did not ordinate the letters according the mean value; this way does not seem to be unusual, to our knowledge and according to Piepho, H. P. (2018). Letters in mean comparisons: what they do and don't mean. Agronomy Journal 110(2), 431-434. The significant differences in the means for forests and grasslands come from the large number of samples in each category (respectively n=526 and n=481). We verified all the panels as you suggested and all are correct. As an example, the detailed result of the Wilcoxon test in R for the T90_HC_PYR is as follow:

```
Pairwise comparisons using Wilcoxon rank sum test with continuity
correction data: Smalldata$T90_HC_PYR and Smalldata$nom_occupation

                       croplands      forests      grasslands

forests                < 2e-16          -              -

grasslands             < 2e-16        1.4e-07          -

vineyards & orchards   1.6e-08        < 2e-16        < 2e-16

P value adjustment method: holm
```

**Edit:** No modification needed.

*Figures 4 and 5: please consider combining the two figures and replacing the histograms in Fig 4 with gradient bars. I found myself wanting to compare land use with the indicators and would prefer them to be adjacent.*

**Answer:** The figure will be modified according to your suggestion in the revised version.

**Edit:** The map on former Fig. 5 has been combined with Fig. 4. The histograms have been replaced by gradient bars. Captions and references to the figures in the paper have been modified accordingly.

**Reply on Topical Editor**

*Dear Authors,*

*Overall, this is a well-written manuscript on an extensive and valuable dataset. Thank you for your thorough responses to the reviewer comments, which are satisfactory and, if followed, will result in an improved manuscript. I have some additional minor suggestions below. Please provide point-by-point responses to the below and to Reviewer 2's comments with line references to corresponding changes in the manuscript.*

*Best wishes,*
*Jocelyn Lavallee*

**We thank you for accepting to be our Topical Editor and for your review and comments.**

*Line 130: As this is the first mention of R-index and I-index, it would be good to define them here (or refer to their definition in the next section)*

**Edit:** We referred to their definition in 2.3.2.
*Lines 156, 158, 160, 161: change "have" to "has"*

**Edit:** Corrected.

*Line 189: Please explain the choice to use no intercept in the linear models and provide a citation if applicable*

**Edit:** The intercepts for the different panels a, b, c, d were respectively -0.62, -0.71, 1.66, -0.81. Additionally, the analysis of several empty Rock-Eval® pods only showed a very weak signal (TOCre6 < 0.2 gC.kg$^{-1}$).

*Line 199: missing a closing parenthesis*

**Edit:** The closing parenthesis was in former L.200 to include the description of the R packages.

*Line 299: change "plain" to "plains"*

**Edit:** The plural is attached to areas ("plain areas").

*Line 387: I agree with Reviewer 1 that "mess up" could be changed to something like "negate", "dampen", "weaken", etc.*

**Edit:** We replaced "mess up" by "dampen" as suggested.

*Conclusion: you may consider adding a point about the potential to use Rock-Eval to quantify inorganic, as well as organic, SC since it was a key point in the results (line 335)*

**Edit:** We added a sentence to highlight the potential of Rock-Eval® thermal analysis in this regard.

*Figure 1 (and throughout manuscript): Please keep naming of variables consistent for clarity. In panel (c), the x axis label (InoC) is not consistent with the caption (Cinorg). In panel (d), the axis labels are not consistent with the caption (e.g., "Total C" vs. TOCre6+Minc"). Please also capitalize the C in "Minc".*

**Edit:** Both the figure and the caption have been corrected. We also corrected C_N to C/N, S2_PC to S2/PC, PC_TOCre6 to PC/TOCre6, HI_OIre6 to HI/OIre6, MINC to MinC, R_index to R-index and I_index to I-index in Table B. A final review allowed us to correct a few other spelling mistakes and typos.

*Figure 3: Given the many acronyms in the manuscript, I suggest writing out "hydrogen index" in the figure caption, and ideally the axes, for added clarity.*

**Edit:** Both the figure and the caption have been corrected.